# Mapping and Characterization of HCMV-Specific Unconventional HLA-E-Restricted CD8 T Cell Populations and Associated NK and T Cell Responses Using HLA/Peptide Tetramers and Spectral Flow Cytometry

**DOI:** 10.3390/ijms23010263

**Published:** 2021-12-27

**Authors:** Amélie Rousselière, Laurence Delbos, Céline Bressollette, Maïlys Berthaume, Béatrice Charreau

**Affiliations:** INSERM, Center for Research in Transplantation and Translational Immunology, Nantes Université, UMR 1064, CHU Nantes, F-44000 Nantes, France; amelie.rousseliere@etu.univ-nantes.fr (A.R.); Laurence.Delbos@univ-nantes.fr (L.D.); celine.bressollette@chu-nantes.fr (C.B.); mailys.berthaume@etu.univ-nantes.fr (M.B.)

**Keywords:** HLA-E, CD8 T cells, NK, γδT, HCMV, pHLA tetramers, spectral flow cytometry

## Abstract

HCMV drives complex and multiple cellular immune responses, which causes a persistent immune imprint in hosts. This study aimed to achieve both a quantitative determination of the frequency for various anti-HCMV immune cell subsets, including CD8 T, γδT, NK cells, and a qualitative analysis of their phenotype. To map the various anti-HCMV cellular responses, we used a combination of three HLA_peptide_ tetramer complexes (HLA-E_VMAPRTLIL_, HLA-E_VMAPRSLLL_, and HLA-A2_NLVPMVATV_) and antibodies for 18 surface markers (CD3, CD4, CD8, CD16, CD19, CD45RA, CD56, CD57, CD158, NKG2A, NKG2C, CCR7, TCRγδ, TCRγδ2, CX3CR1, KLRG1, 2B4, and PD-1) in a 20-color spectral flow cytometry analysis. This immunostaining protocol was applied to PBMCs isolated from HCMV^−^ and HCMV^+^ individuals. Our workflow allows the efficient determination of events featuring HCMV infection such as CD4/CD8 ratio, CD8 inflation and differentiation, HCMV peptide-specific HLA-E_UL40_ and HLA-A2_pp65_CD8 T cells, and expansion of γδT and NK subsets including δ2^−^γT and memory-like NKG2C^+^CD57^+^ NK cells. Each subset can be further characterized by the expression of 2B4, PD-1, KLRG1, CD45RA, CCR7, CD158, and NKG2A to achieve a fine-tuned mapping of HCMV immune responses. This assay should be useful for the analysis and monitoring of T-and NK cell responses to HCMV infection or vaccines.

## 1. Introduction

Human cytomegalovirus (HCMV; human herpesvirus 5, HHV5) is the prototype member of *β-herpesvirus* family and a widespread opportunistic pathogen. In healthy individuals, primary infection is subclinical and is followed by a life-long, persistent infection that is controlled by host immune system [1]. However, HCMV is a major cause of morbidity and mortality in immunocompromised individuals such as transplant recipients and patients with HIV infection. Immune response against HCMV is complex, multifactorial, and includes a set of persistent and virus-specific effector NK and CD8 αβT and γδT cell populations [2,3,4]. These effector cells display cytotoxic functions devoted to eliminating infected cells and preventing further HCMV reactivation [5,6]. HCMV-reactive CD8 αβT cells against viral peptides (pp65, IE1, etc.) presented by conventional MHC class I (HLA-A and HLA-B) molecules have been well characterized [7]. These conventional CD8 T cell responses are usually associated with an efficient control of infection [5]. In addition, CD8 T cell responses bearing αβTCR but recognizing non-classical MHC, and HLA-E molecules presenting peptides derived from HCMV UL40 protein have emerged as non-conventional T cell responses, also observed in HCMV seropositif (HCMV^+^) hosts including transplant recipients and healthy individuals [8,9,10]. In a previous study, we have detected HLA-E-restricted CD8 T targeting UL40 peptides (HLA-E_UL40_ CD8 T) in more than 30% of HCMV^+^ kidney transplant recipients. We have shown that HLA-E_UL40_ CD8 T cell responses may represent up to 30% of total blood CD8 T cells in a host post-infection [11].

In contrast to conventional HLA class I-restricted anti-HCMV CD8T cells, the frequency, the peptide-specificity, and the function of HLA-E_UL40_CD8 T cell responses remain mostly unknown [12]. Although the first pieces of knowledge have been reported on the phenotype and functions of HLA-E_UL40_ CD8 T cells, a broader characterization of their phenotype in comparison with other, conventional anti-HCMV HLA-I-restricted CD8 T such as HLA-A*02pp65 and HLA-A*02 IE1 cells remains to be established. Concerning their functions, previous studies reported that HLA-E_UL40_ CD8 T cells can be activated in vitro and display cytotoxic activity against cellular targets, such as endothelial cells, expressing HLA-E molecules loaded with UL40 peptides [11,12,13]. The specific implication of anti-HCMV HLA-E_UL40_ CD8 T cells to the immune control of HCMV infection is still unknown. It can be speculated that, similar to other HCMV-specific CD8 T cells, HLA-E-restricted cytotoxic T cells (CTL) patrol to detect and eliminate infected cells. Another non-exclusive function for HLA-E-restricted CTL could be to ensure the regulation of other anti-HCMV cellular responses such as NK or γδ T or CD4T cells. Importantly, due to a full sequence homology between the UL40 peptides frequently provided by HCMV strains and some HLA class I peptides, both presented by HLA-E, it has been established that anti-HCMV HLA-E_UL40_ CD8 T cell responses may cross-react with allogeneic HLA in the setting of solid organ transplantation [11]. Whether this cross-recognition of donor HLA-E_HLA-I_ on allograft could mediate transplant injury and rejection remains to be determined. Nevertheless, the biological relevance of this persistent CD8 T cell subset in the course of HCMV infection and recovery is still unknown.

Detection and analysis of HCMV antigen-specific CD8 T cells require the use of HLA monomers loaded with viral peptides. The interaction between the peptide-HLA (pHLA) complex and a cognate TCR is short lived and of low affinity. By way of multimerization of four peptide-HLA molecules on a streptavidin scaffold, tetramers increase the avidity of the pHLA:TCR interaction by engaging several of the TCRs expressed on a specific T cell, thereby stabilizing the interaction [14]. This allows for direct and specific staining of the T cells. pHLA tetramers provide a simple, fast, and efficient approach for monitoring and handling specific T cells in patients. Peptide-HLA tetramers can be used to identify specific T cells without further in vitro manipulation, and they allow for a simultaneous evaluation of the differentiation state through co-staining for various cell surface markers for differentiation, activation, and exhaustion [15,16,17]. Spectral flow cytometry measures the complete emission spectrum to identify fluorochromes. Consequently, in contrast to conventional flow cytometry, spectral flow cytometry is able to differentiate fluorochromes with significant overlap in the emission spectra, enabling the use of spectrally similar fluorochrome pairs in a single assay and thus increasing the number of protein markers that can be analyzed concomitantly [18,19]. The present study presents the set up and validation of a method based on the use of a set of HLA-A_peptide_ and HLA-E_peptide_ complexes in combination with antibodies for 20 cell markers for the concomitant detection, quantification, and immunophenotyping of non-conventional anti-HCMV HLA-E_UL40_ CD8 T cells, conventional HLA-A*0201_pp65_ and other immune responses including NK and γδT cell subsets regulated upon HCMV infection.

## 2. Results

### 2.1. Quantitative Assessment of CD8 αβT Cells in Response to HCMV Infection: CD8 Inflation and Frequencies of Anti-HCMV Peptide-Specific CD8 T Cells

Here, to define our ability to map the various anti-HCMV cellular responses post-infection, we used a combination of three pHLA tetramer complexes (HLA-E_VMAPRTLIL_, HLA-E_VMAPRSLLL_, HLA-A2_NLVPMVATV_) and antibodies for 18 surface markers (CD3, CD4, CD8, CD16, CD19, CD45RA, CD56, CD57, CD158, NKG2A, NKG2C, CCR7, TCRγδ, TCRγδ2, CX3CR1, KLRG1, 2B4, and PD-1) in a 20-color multiparameter flow cytometry analysis. For validation, our immunostaining protocol was applied to PBMCs isolated from HCMV- (*n* = 4) and HCMV+ (*n* = 4) individuals and cytometry data were analyzed post-acquisition as follows.

#### 2.1.1. Lymphocyte Gating

As an initial gating strategy to discriminate anti-HCMV NK and T cell populations (Figure 1), both forward and side scatter (FSC-Area (A) vs. FSC- Height (H) and SSC-H vs. SSC-A) dot plots were used to exclude doublets and to target singlets only (Figure 1A). Next, FSC vs. SSC gating was used to identify lymphocytes based on size and granularity. It is often suggested that forward scatter indicates cell size, whereas side scatter relates to the complexity or granularity of the cell. This gating strategy is also used to exclude debris, as they tend to have lower forward scatter levels. They are found at the bottom left corner of the FSC vs. SSC density plot. Next, live cells were selected using Fixable Viability Stain 440UV as a viability marker. This dye reacts with and covalently binds to cell-surface and intracellular amines. Permeable plasma cell membranes, such as those present in necrotic cells, allow for the intracellular diffusion of the dye and covalent binding to higher overall concentrations of amines than in non-permeable live cells. Using the expression for CD3 and γδTCR allows us to determine three lymphocyte subsets: CD3^−^ γδTCR ^−^ cells including mostly B lymphocytes and NK cells, CD3^+^ γδTCR^−^ cells including mostly αβTCR T cells, and finally, CD3^+^ γδTCR ^+^ cells, which include γδT cells (Figure 1A). Thus CD3/ γδTCR costaining provides the distribution of these three subsets among PBMC samples issued from HCMV^+^ patients and controls and may be indicative of γδTCR inflation post-infection.

#### 2.1.2. CD4/CD8 Ratio

CD4 and CD8 expression was then examined in the CD3^+^ γδTCR^−^ lymphocyte population to define the percentage of CD4^+^ T cells, CD8^+^ T cells as well as double positive and negative CD3^+^ T cells (Figure 1B). As illustrated in Figure 1C, our data show that HCMV- individuals display a higher percentage of CD4 compared to CD8 (mean values: 66.6% vs. 33.4% for CD4 and CD8 T cells, respectively, from HCMV- individuals (*n* = 4), *p* < 0.05). In contrast, HCMV^+^ individuals, even at a distance from primary infection, display no significant difference in the percentages of CD4 versus CD8 T cells (mean values: 48.9% vs. 51.1% for CD4 and CD8 T cells, respectively, from four HCMV^+^ individuals), indicative of an HCMV-induced CD8 T cell inflation. Indeed, expansion of the CD8 T cell pool occurs early post-infection and is a hallmark of HCMV and HIV infections [20,21,22]. 

#### 2.1.3. HCMV Peptide-Specific CD8 T Cell Responses

Three pHLA tetramer complexes were used in conjunction with anti-CD8 antibodies for the immunostaining of conventional and unconventional HCMV peptide-specific CD8 T cell responses. Conventional CD8 T cells were detected using HLA-A*0201(A2)/_pp65_ (NLVPMVATV) tetramer complexes, while unconventional HLA-E restricted CD8 T cells were detected using two HLA-E tetramer complexes containing two different UL40 signal peptides (HLA-E/_VMAPRTLIL_ and HLA-E/_VMAPRSLLL_). Examples of detection using HLA-A2_pp65_ and HLA-E_UL40_ tetramers and CD8 costaining are shown in Figure 1D,E and reveal similar frequency (from 2 to 6% of total CD8 T cells) for both HCMV antigen-specific, HLA-A2_pp65_ and HLA-E_UL40,_ CD8 T cells, consistent with our previous studies [11] Thus, CD8/HLA classI/peptide tetramer costaining allows the detection and quantification of HCMV-specific conventional but also unconventional, HLA-E-restricted CD8 T cell populations using spectral flow cytometry. 

### 2.2. Immunophenotyping of CD8 T Cell Responses

#### 2.2.1. CD8 T Cell Differentiation

HCMV peptide-specific CD8 T cells stained with pMHC class I tetramers can be further characterized by immunophenotyping using antibodies for CD45RA, CCR7, CX3CR1, PD-1, CD56, CD57, CD158, NKG2A, NKG2C, KLRG1, and 2B4. Firstly, costaining for CD45RA and CCR7 allows us to segregate CD8 T cells according to their differentiation state: CD45RA^+^ CCR7^+^ CD8 T cells are defined as naive T cells (TN), CD45RA^−^ CCR7^−^ as central memory T cells (TCM), CD45RA^−^CCR7^+^ as effector memory T cells (TEM), and CD45RA^+^CCR7^−^ as terminally differentiated T cells re-expressing CD45RA (TEMRA) (Figure 2A) [23]. Consequently, CD45RA/CCR7 costaining enables a comparative analysis of CD8 T cell differentiation for HCMV peptide-specific CD8 T cells stained with pHLA class I tetramers, such as conventional vs. unconventional (HLA-E-restricted) CD8 T and a comparison between HCMV peptide-specific CD8 T cells and total (tetramer negative) CD8 T cell pool (Figure 2A). As illustrated in Figure 2A lower panel, and consistent with previous studies [22,23], a majority of CD8 T cells (around 70%) in HCMV^+^ individuals are TEMRA cells expressing CD45RA but not CCR7. Considering HCMV-specific responses, only a part of HLA-A2_pp65_ and almost all anti-HCMV HLA-E_UL40_ CD8 T cells stained with HLA-E_UL40_ tetramers that persist in HCMV^+^ individuals post-infection display a CD45RA^+^ CCR7^−^ phenotype and thus belong to TEMRA cells (Figure 2A). Figure 2B provides a quantification of the CD8 differentiation stages in HCMV^+^ versus HCMV^−^ individuals, indicating a trend toward less TN and more TEMRA in HCMV^+^ individuals, which may reflect the impact of HCMV-specific CD8 populations, as previously reported [22,24]. 

#### 2.2.2. Immunophenotyping of Naive, Central Memory, Effector Memory, and Terminally Differentiated CD8^+^ T Cell Subsets

Concomitant costaining with a panel of antibodies was performed to investigate, in a single assay, receptors for T cell activation and inhibition (2B4, PD-1, CD158, NKG2A, NKG2C, and KLRG1), migration (CX3CR1), and cytotoxic and proliferation capacity (CD56, CD57). This antibody panel was used to better characterize and to compare the various T cell subsets according to their differentiation state, as illustrated in Figure 2C. CD8 T cell differentiation from TN to TEMRA is associated with gain and loss of expression for several receptors, as previously reported [23]. Upon differentiation, CD8 T cells acquire both CX3CR1 and the inhibiting receptor KLRG1, which are not expressed on TN but appear on TEM and are coexpressed (80% of cells) on TEMRA. Similarly, the expression of the activating receptor 2B4 progress along differentiation with majority of TEMRA (55%) being 2B4^+^. NKG2C and CD158 are expressed or even coexpressed on TEMRA only. Concerning cytotoxic activity, the expression of CD56 is null for CD8 TN and TCM, appears maximal for TEM (18.2%), and then decreases for TEMRA (5.2%), while CD57 appears on TEM (13.7%) and further increases for TEMRA (27.8%). TEM and TEMRA, coexpressing both CD56 and CD57, represent 2.4% and 12.0%, respectively. Only a small portion of CD8 TEMRA express the activating receptor NKG2C (4.84%, and among them, 1.39% coexpress CD158). These data indicate that our workflow is robust enough to provide an accurate phenotype comparison across the four differentiation stages of CD8 T cells, thus enabling us to characterize HCMV-specific CD8 T cell subsets. 

### 2.3. Deciphering γδT and Vδ2^−^γδT Cells upon HCMV Infection

The γδ T cells are an integral part of the immune response against HCMV [6,25]. Using our protocol, the use of anti-γδTCR antibodies allows the positive selection of lymphocytes expressing both CD3 and an γδTCR, thus excluding conventional CD3^+^ T lymphocytes that bear conventional αβTCR (Figure 3A). This gating step provides a quantification for total γδT cells in blood samples from HCMV^−^ and HCMV^+^ hosts (Figure 3A,B). Consistent with previous studies [6], γδT cells comprise around 10 ± 7% of total CD3^+^ cells in both HCMV- and HCMV+ individuals (Figure 3B). In a subsequent step, subgating of CD3^+^ γδT according to the expression of δ2TCR chain and CD8 provided a mean to focus on CD3^+^ γVδ2^−^ γδ T cells. In humans, γδ T cells are divided in two subsets, the Vγ9^+^Vδ2^+^T cells that are found predominantly in the blood and all the other γδ T cells (collectively called Vδ2^−^γδ T cells, and mainly composed of Vδ1^+^ and Vδ3^+^ T cells) that are primarily located in tissues, particularly in epithelia [6]. HCMV infection leads to a strong increase (in proportion and number) in γδ T cell subsets in the blood circulation, which persisted long term [26]. HCMV induces the expansion of Vδ2^−^γδ T cells in the blood, which correlates with the resolution of infection providing evidence for an antiviral function of these subset of γδ T cells [27,28]. Our data illustrate the predominance of Vδ2^−^γδ T cells over Vδ2^+^γδ T cells in HCMV^+^ hosts with a frequency that ranges between 38% and 97% (mean value: 65.6%) of total CD3^+^ γδT cells but with large individual variations (Figure 3C). HCMV-induced γδ T cells mostly express an effector/memory TEMRA phenotype (Figure 3A) with similarities to the one described for HCMV-specific CD8^+^αβ T cells [23]. Subgating on CD45RA/CCR7 indicated divergent differentiation status for Vδ2^−^γδ T cells vs. Vδ2^+^γδ T cells with almost all Vδ2^−^γδ T beeing TEMRA (CD45RA^+^CCR7^−^), whereas Vδ2^+^γδ T cells include mostly less-differentiated T cells in HCMV^+^ individuals. To further characterize HCMV-induced γδ T cells, the phenotype of both subsets was investigated for the immune receptors used for phenotyping CD8 αβT cells. Comparison of expression pattern further highlights phenotype differences between Vδ2^−^ and Vδ2^+^ γδ T cells and suggests that coexpression for CX3CR1 and KLRG1 and CD56/CD16 expression featured Vδ2^−^ γδ T cells (Figure 3A).

### 2.4. Analysis of HCMV-Induced NK Cell Subsets

In healthy human adults, NK cells comprise 5–15% of circulating lymphocytes; together with T cells and B cells, they are one of the three major lymphoid lineages [29]. Within lymphocytes, NK cells are phenotypically defined as CD56^+^ cells that do not express T (CD3) or B (CD19) cell lineage markers. In our protocol, sequential gating on CD3^−^TCR γδ ^−^ cells followed by the exclusion of CD19^+^ cells allowed us to define CD3^−^CD19^−^CD56^+/-^ as NK cells (Figure 4A,B). Accordingly, the total percentages of NK cells among peripheral lymphocytes were calculated in samples from HCMV^−^ and HCMV^+^ individuals and are shown in (Figure 4C). The expression of CD56 in combination with CD16, the low-affinity Fc γ receptor IIIa, further allows to distinguish different NK cell subsets [30,31]. By examining CD56 and CD16 costaining, we were able to identify NK cells at different stages of differentiation, including the immature CD56^−^/CD16^−^, the early differentiated CD56^bright^CD16^+/−^, the mature CD56^dim/bright^CD16^+^, and the terminally differentiated CD56^−^CD16^+^ NK cell subsets [32,33]. When comparing the frequency of each subset in HCMV^−^ and HCMV^+^ individuals, we found that CD56^dim^CD16^+^ respresent the vast majority of NK cells in both groups (Figure 4D). HCMV infection triggers the specific expansion of mature CD56^dim^CD16^+^ NK, expressing the CD94/NKG2C activating receptor and coexpressing the CD57 with a high cytotoxic activity. Thus, subgating using CD57 and NKG2C markers was performed for all the samples to calculate the percentages of mature CD56^dim^CD16^+^ CD57^+^ NKG2C+ in HCMV^−^ and HCMV^+^ hosts, as shown in Figure 4E. Our data indicated that, although no statistically significant difference was achieved due to individual variability, the frequency of mature, memory-like CD56^dim^CD16^+^ CD57^+^ NKG2C^+^ appears higher in HCMV^+^ compared to HCMV^−^ hosts.

## 3. Discussion

Flow cytometry provides a high-throughput and cost-effective method of immunophenotyping and immunomonitoring of patients (currently, more than 40 fluorophores are available) on many cells with high-throughput (approx. 10,000 events/s). In contrast to flow cytometry, which uses fluorescent molecules, mass cytometry uses heavy metal tags and time-of-flight mass spectrometry readouts to measure antibody binding to cells [34]. This method allows a much larger number of simultaneous markers than conventional flow cytometry. Mass cytometry, on the other hand, acquires cells at a much lower rate (approx. 300–400 events/s) but with more markers per cell (over 50). Spectral cytometry improves conventional flow cytometry by increasing the number and combination of fluorophores, thereby providing increased flexibility of panel design, as well as incorporating autofluorescence measurement and extraction [19]. By the use of fluorophore-conjugated antibodies, staining, and analysis protocols already established for conventional cytometry, spectral cytometry provides a readily accessible technique [35].

Here, we report on the development of a staining protocol and a staining strategy combining HLA class I/HCMV peptides tetramer complexes and a panel of 18 antibodies to study HCMV-specific immune cell responses. HLA/peptide tetramer staining offers the possibility to detect and quantitate peptide-specific anti-HCMV CD8 T cell populations [16]. The tetramers that we used include HLA-A2pp65 tetramers and HLA-E_UL40_ tetramers to decipher HLA-E-restricted CD8 T cells induced in the response to HCMV infection. It is important to emphasize that HLA-E_UL40_ tetramers bind to both NK and T cells. HLA-E is a ligand for the heterodimeric CD94/NKG2A/C receptors [36], which are expressed at high level on NK cells and are also expressed at lower level on CD8 T cells [36,37]. To allow a TCR-specific binding and to avoid the binding of HLA-E_UL40_ tetramers to CD94/NKG2A and CD94/NKG2C receptors, we performed a CD94 blockade using blocking antibodies as a preliminary step of immunostaining as we previously reported [11,13]. A major result from this study was the efficient detection of both HLA-A2_pp65_ and HLA-E_UL40_ CD8 T cells stained with the tetramers in our experimental conditions using spectral cytometry. In the present study, we found that the frequency of pp65 and UL40 epitope-specific T cells among CD8 T cells was in the range of the frequency that we previously reported with conventional flow cytometry [11]. This result indicates that the binding of pHLA tetramer remains stable during the processes of immunostaining and data acquisition and is strong enough to allow cell detection by spectral flow cytometry. Therefore, pHLA-E tetramers against UL40 HCMV epitopes combined with cell surface markers allow us to study these HCMV-specific CD8 T cell responses in more detail in a large cohort of patients. Previous studies established the emergence of HLA-E-restricted CD8 T cell subsets in autoimmune [38] and infectious diseases [12,39]. Functionally, in some studies, HLA-E-restricted CD8 T cells have been shown to display cytotoxic activities [8,11,39] toward autologous, allogeneic, or infected cells expressing HLA-E such as endothelial cells [40] but also regulatory functions [38], as reported in mice [41]. The role of HLA-E-restricted CD8 T cells in the outcome on HCMV infection is still unknown. Previous analysis of phenotype identified HCMV-specific HLA-E CD8 T as terminally differentiated TEMRA cells expressing CD56 [8,11]. Commonly, there is no known specific surface receptor that leads to HLA-E_UL40_ CD8 T cell identification within PBMCs that may help to analyze or sort these T cells without using HLA-E tetramers. The phenotypic and molecular characteristics of these CD8+ T cells therefore require further study. To investigate further the role that HLA-E_UL40_ CD8 T cells may play in the control of HCMV infection, we sought to set up an integrated analysis of the multiple cellular responses induced by the infection. Our panel of antibodies enables the concomitant determination in a single sample of the frequency for a set of anti-HCMV responses such as conventional and unconventional peptide-specific CD8 T cells, total γδT and δ2^−^γT cells, immature and mature NK, and memory-like NK cells expressing NKG2C and/or CD57 [6,42]. Each subset can be further characterized for the expression of several markers including 2B4, PD-1, KLRG1, CD45RA, CCR7, CD158, and NKG2A to achieve a fine-tuned analysis of HCMV immune responses. Future applications for this assay include a better knowledge of HCMV infection through the comprehensive analysis of NK and T cell responses to HCMV infection or vaccines and a tool for the stratification of transplanted patients according to risk factors related to HCMV infection [43,44].

## 4. Materials and Methods

### 4.1. Samples and Reagents

Blood samples collected from seronegative (HCMV^−^) and seropositive (HCMV^+^) anonymous healthy volunteers (HV) were obtained from the Etablissement Français du Sang (EFS des Pays de La Loire, Nantes, France) with donors’ specific and written informed consent for research use. PBMCs were isolated by Ficoll density gradient (Eurobio, Les Ulis, France) and keep frozen until used. Banked biological samples (PBMCs) from HCMV^+^ kidney transplant recipients were issued from the DIVAT biocollection (CNIL agreement n°891735, French Health Minister Project n°02G55). PBMCs from patients who underwent kidney transplantation in the Institute for Transplantation Urology Nephrology (ITUN, CHU de Nantes, France) were prospectively isolated from blood samples, frozen, and stored at the Centre de Ressources Biologiques (CRB, CHU de Nantes, France). PBMCs were thawed before use in RPMI-1640 medium (Gibco, Amarillo, TX, USA) supplemented with 10% human serum (Gibco), 2 mM L-glutamine (Gibco), 100 U/mL penicillin (Gibco), and 0.1 mg/mL streptomycin (Gibco). To set up the present protocol, we used blood samples that we previously tested for the absence or the presence of HCMV peptide-specific CD8 T cells using a set of pHLA tetramers and conventional flow cytometry. Four PBMC samples containing different HCMV peptide-specific CD8 T cells were then selected for the study. These 4 samples were issued from 2 HCMV^+^ healthy donors and from 2 HCMV^+^ kidney transplant recipients. The two groups included 2F/2M and 1F/3M for HCMV^−^ and HCMV+ individuals, respectively. The mean values of ages are 47.5 ± 14.7 and 55 ± 14.7 years for HCMV- and HCMV+, respectively, and thus are not significantly different.

### 4.2. HLA-E_UL40_ and HLA-A*02:01_pp65_ Tetramer Complexes

Peptides from HCMV UL40 protein (AA_15-23_: VMAPRTLIL and VMAPRSLLL) and the UL83 (pp65) protein (AA_495-503_: NLVPMVATV) were synthesized (purity>95%) and purchased from Genecust (Boynes, France). The HLA_peptide_ monomers HLA-E*01:01_UL40_ and HLA-A*02:01_pp65_ were produced by the recombinant protein core facilities (P2R, SFR Bonamy, Université de Nantes, France) as we previously reported [11]. HLA_peptide_ monomers were biotinylated, purified, and tetramerized using APC-streptavidin (BD Biosciences, Le Pont de Claix, France).

### 4.3. Spectral Flow Cytometry: Immunostaining, Acquisition and Post-Acquisition Data Analysis

The 20-marker panel was optimized for use on a Cytek Aurora (Cytek Biosciences, Fremont, CA, USA) spectral flow cytometry platform with a 5-laser configuration (laser excitation wavelengths: 355 nm, 405 nm, 488 nm, 561 nm, and 640 nm). Before use, titration experiments were carried out to determine the antibody concentration providing highest staining index. For immunophenotyping, cells (1.10^6^ PBMCs/well in 96-well plates) were costained using a multistep protocol. PBMCs were washed twice in RPMI and cells and filtered through a 100 μm filter (ThermoFisher, Waltham, MA USA) before immunostaining. PBMCs were then incubated with a viability marker, Fixable Viability Stain 440UV (BD Bioscience) and diluted in PBS (100 µL) for 15 min at 4 °C. PBMCs were washed twice in PBS and centrifugated at 2500 rpm for 2 min at 4 °C. Next, cells were incubated with anti-NKG2A (Biotechne, Noyal-Châtillon-sur-Seiche, France) and anti-NKG2C (BD Bioscience) mAbs for 15 min at 4 °C, diluted in PBS (30 µL) before being incubated in PBS (30µL) for 25 min at 4 °C with blocking anti-CD94 mAb (BD Bioscience) to avoid the binding of HLA-E_UL40_ tetramer to CD94/NKG2A/C receptors. After a washing step, PBMCs were incubated with APC-labeled -HLA_peptide_ tetramers (50µg/mL in 30µL PBS) for 20 min at RT. After washing, PBMCs were incubated successively for 10 min at 4 °C with 5 cocktails of antibodies diluted in PBS (30µL): cocktail 1 containing Fc-block™ reagent (BD Bioscience) and anti-CCR7, cocktail 2 containing anti-TCRγδ and anti-TCRγδ2, cocktail 3 containing anti-CX3CR1 mAbs alone, cocktail 4 containing anti-CD158, -KLRG1, -2B4, -PD-1 mAbs, and cocktail 5 containing anti- CD4, -CD57, -CD3, -CD45RA, -CD8, -CD56 and -CD16 mAbs. After a washing step, PBMCs were finally incubated with anti-CD19 mAbs for 15 min at 4 °C. All antibodies are listed in Table 1. PBMCs were washed twice and were resuspended in PBS before fluorescence analysis. For our study, a mean value of 40,000 events (viable lymphocytes)/samples were acquired for analysis. The fluorescence intensities were measured with a five-laser Cytek Aurora™ spectral flow cytometer (Cytek Biosciences) using SpectroFlo™ software version 2.2.0 (Cytek Biosciences). Using online fluorescence spectra viewers, we were able to identify 20 fluorophores with distinct signatures that could be used in the panel. The selected fluorophores included BUV395, UV440, BUV496, BUV563, BUV737, BUV805, BV421, VioBlue, BV510, BV570, BV605, BV785, FITC, PerCPeFluor710, PE, AlexaFluor594, PE-Cy7, SparkNir685, APC Fire750, and APC (Table 1). The spectral profile of unstained cells was collected and treated as an independent parameter, which allows the autofluorescence signature to be extracted using the unmixing algorithm. The full emission spectrum of each single-stained sample was performed using compensation beads (OneComp eBeads™, Thermo Fisher) or PBMCs and was used to determine the contribution of each fluorophore in a mixed sample using spectral deconvolution (unmixing) algorithms before experiments. The fluorophore spectral signatures obtained at the cytometer were compared to the gold standard full-spectrum signatures shown in the Aurora fluorochrome guide (https://cytekbio.com/blogs/resources/5l-full-spectrum-cytometry-overview-poster, accessed on 1 November 2021) to ensure fluorophore identity and quality. Post-acquisition, unmixed FCS files were conventionally compensated before the data analysis. The frequency of major immune cell populations was determined using FlowJo™ Software v10 (BD Biosciences) based on manual gating strategies as reported in the results section. 

### 4.4. Statistical Analysis

Comparisons between groups were represented as box plots showing median, 25th, and 75th percentile values using GraphPad Prism 8.0 software (GraphPad Software Inc., Sand Diego, CA, USA). Comparisons among groups were performed using non-parametric Wilcoxon-Mann–Whitney tests when suitable. Statistical differences were determined by GraphPad Prism 8.0. A two-sided *p* value < 0.05 was considered to be statistically significant. *p* value: * for *p* < 0.05.

## Figures and Tables

**Figure 1 ijms-23-00263-f001:**
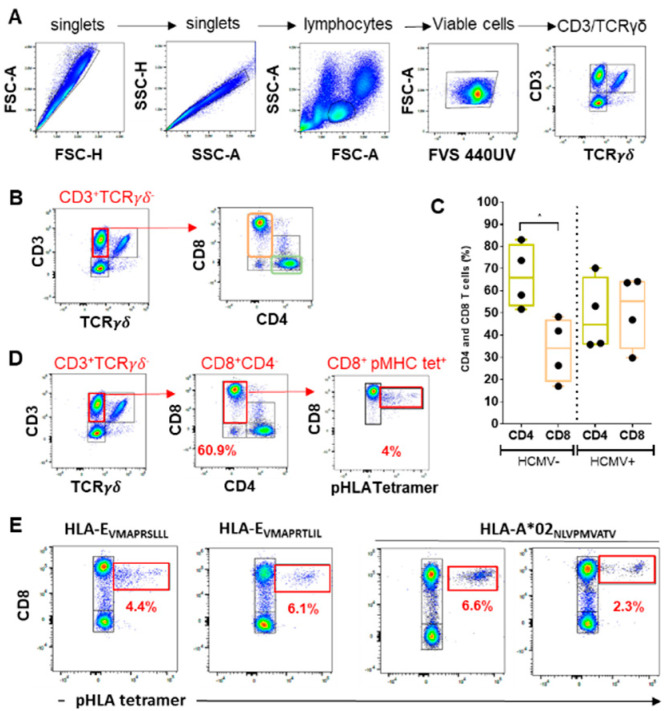
(**A**) A representative nested gating strategy illustrating lymphocyte population subgated by the expression of CD3 and γδTCR. Cells were gated first on FSC-A vs. FSC-H plots and then on SSC-H vs. SSC-A plots to eliminate doublets. Lymphocytes were gated on an SSC-A vs. FSC-A dot plot. Viable lymphocytes were selected using fixable viability stain (FVS) 440UV staining. Lymphocytes were subgated using CD3 and γδTCR staining. (**B**,**C**) Analysis of CD4 and CD8 lymphocyte subsets. A representative density plot showing CD4 and CD8 staining among the CD3^+^ γδTCR- lymphocytes. (**C**) A graphical and statistical analysis of CD4 and CD8 lymphocyte subsets from independent HCMV^−^ (*n* = 4) and HCMV^+^ (*n* = 4) individuals. (**D**,**E**) Detection and quantification of peptide-specific conventional and non-conventional CD8 T cell populations using HLA class I /peptide tetramers (pHLA). Representative detection of anti-HCMV peptide-specific, conventional (HLA-A2_pp65,_ peptide NLVPMVATV) and two unconventional (HLA-E_UL40,_ peptides VMAPRSLLL and VMAPRTLIL), CD8 T cells from three HCMV^+^ hosts is shown. The percentages of tetramer positive cells (tet+) among total CD8 T cells are indicated. FSC-H: Forward scatter height. FSC-A: Forward scatter area. SSC: Side scatter. TCR: T cell receptor. *p* value: * for *p* < 0.05.

**Figure 2 ijms-23-00263-f002:**
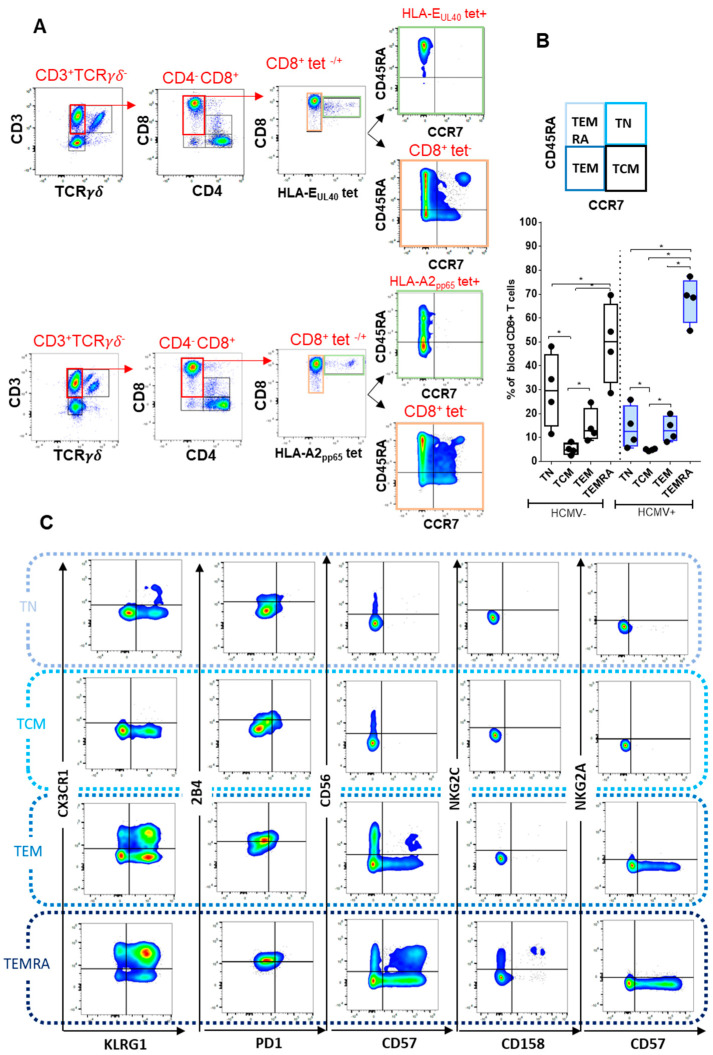
Analysis of CD8 T cell differentiation. (**A**) Representative density plots showing the sequential gating of CD8 T cells negatively (tet-) and positively (tet+) stained by pMHC class I tetramer (HLA-E_UL40 VMAPTSLLL,_upper panel or HLA-A2_pp65_, lower panel) after a selection according to CD4 and CD8 costaining among the CD3^+^ TCRγδ^−^ lymphocytes. The expression pattern for CD45RA/CCR7 of each population (tet^−^ and tet_+_) is shown. (**B**) A schematic representation of CD8 differentiation stages including CD8 naive T cells (TN), central memory T cells (TCM), effector memory T cells (TEM), and terminally differentiated T cells (TEMRA) according to CD45RA and CCR7 expression is indicated (upper panel). A graphical and statistical analysis of differentiation for total CD8 T cell pool from independent HCMV^−^ (n = 4) and HCMV^+^ (*n* = 4) individuals. Data are expressed as box plot with median and interquartile values. (**C**) Immunophenotyping of CD8 T cells for receptors for T cell activation and inhibition (2B4, PD-1, CD158, NKG2A, NKG2C, KLRG1), migration (CX3CR1) and cytotoxic capacity (CD56, CD57)**.** The coexpression of receptors are shown for the 4 differentiation states (TN, TCM, TEM, TEMRA) analyzed for total CD8 T cells and representative of a single HCMV+ individual. *p* value: * for *p* < 0.05.

**Figure 3 ijms-23-00263-f003:**
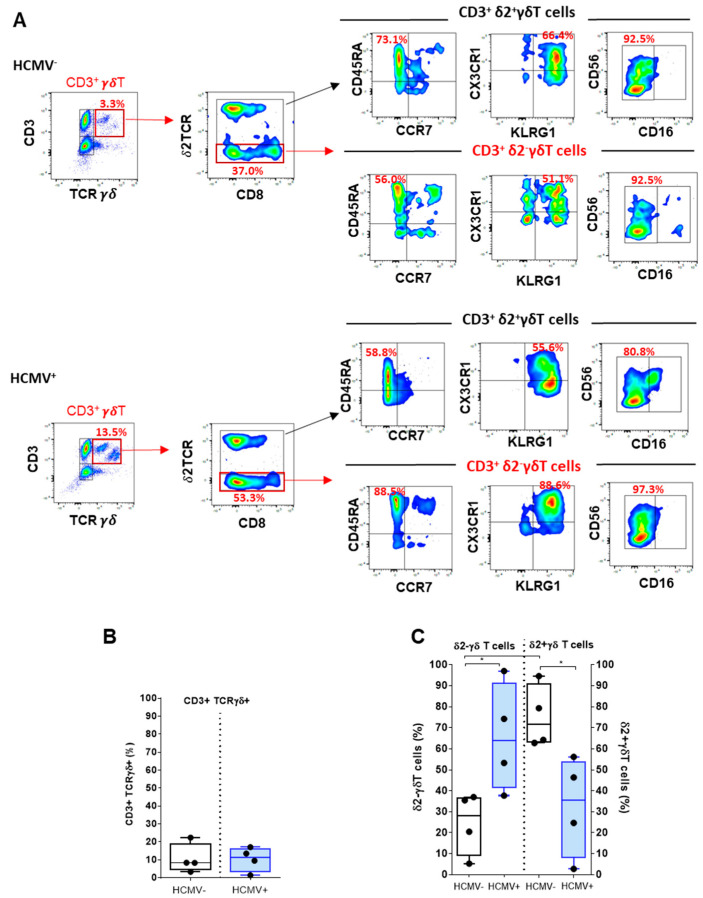
**Analysis of γδT and Vδ2^−^****γδT cells**. (**A**) Representative density plots from a single HCMV^−^(upper panel) and HCMV^+^ (lower panel) individuals showing the sequential selection of CD3^+^ γδT cells using CD3 and γδTCR costaining and of Vδ2^−^γδT and Vδ2^+^γδT subsets using Vδ2γδT and CD8 costaining used to quantify the γδT cell subsets. Immunophenotyping of γδT cell subsets showing costaining for CD45RA/CCR7, CX3CR1/KLRG1 and CD56/CD16. The coexpression of immune receptors for Vδ2^−^γδT vs. Vδ2^+^γδT cell populations are shown. For comparison, cell frequency (%) is indicated for some costainings. (**B,C**) Box plots with median and interquartile values were used to represent the percentages of (**B**) CD3^+^ γδ^+^T among lymphocytes and (**C**) Vδ2^−^γδT vs. Vδ2^+^γδT cells obtained from independent HCMV^−^ (*n* = 4) and HCMV^+^ (*n* = 4) individuals. *p* value: * for *p* < 0.05.

**Figure 4 ijms-23-00263-f004:**
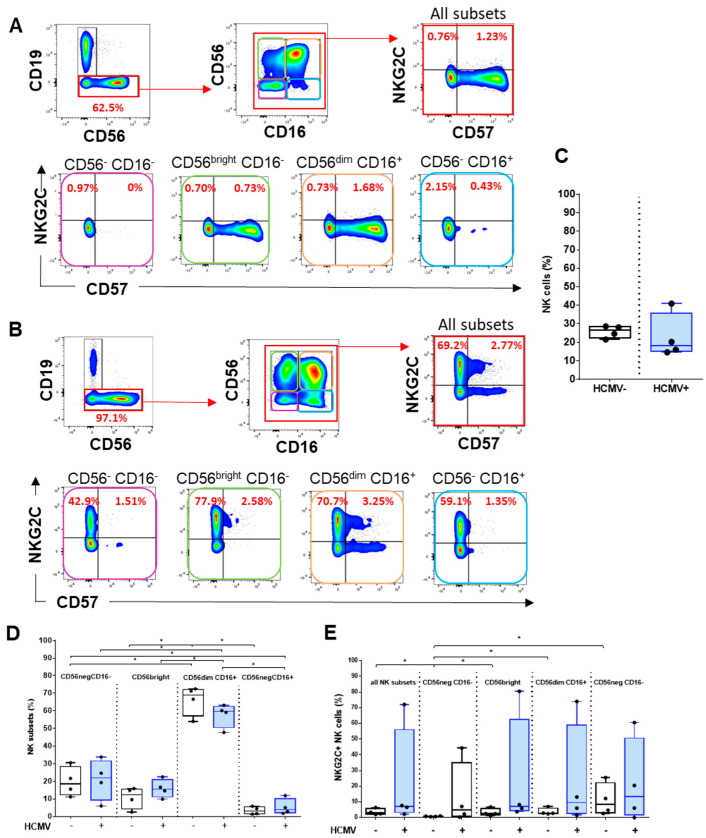
**Analysis of HCMV-induced NK cell subsets**. (**A**,**B**) Representative density plots showing the sequential selection (upper panel, left to right) CD3^−^ γδ^-^T cells, CD19^−^ cells using CD19/CD56 costaining, and CD56/CD16 costaining, enabling the determination of 4 major NK cell subsets. Co-expression for CD57/NKG2C is shown for all NK (upper panel) and for each NK subset (lower panel). The data shown are from one representative HCMV^−^ HV (**A**) and HCMV^+^ kidney transplant recipient (**B**). For comparison, cell frequency (%) is indicated for some costainings. Quantitative analyses are represented as box plots comparing the percentages of total NK cells (**C**), the percentages of NK subsets (**D**), and the percentages of NKG2C^+^ NK cells (**E**) for HCMV^−^ (*n* = 4) and HCMV^+^ (*n* = 4) individuals. *p* value: * for *p* < 0.05.

**Table 1 ijms-23-00263-t001:** Antibody panel and HLApeptide tetramers used for anti-HCMV immune profiling using spectral flow cytometry. Antigen, fluorophore, clone, and dilution are indicated.

Λ Excitation (nm)	Fluorophore	Λ Emission (nm)	Cytometer Detector	Antigen	Antibody Clone	Dilution	Source
355	BUV395	395	UV2	**CD45RA**	5H9	1/80^e^	BD Bioscience
	Fixable Viability Dye UV440	436	UV6	**Viability**	/	1/2000^e^	BD Bioscience
	BUV496	496	UV7	**CD16**	3G8	1/160^e^	BD Bioscience
	BUV563	564	UV9	**NKG2C**	134591	1/20^e^	BD Bioscience
	BUV737	735	UV14	**CD56**	NCAM16.2	1/20^e^	BD Bioscience
	BUV805	803	UV16	**CD8**	SK1	1/80^e^	BD Bioscience
405	BV421	421	V1	**CCR7**	G043H7	1/20^e^	Biolegend
	VioBlue	452	V3	**KLRG1**	REA261	1/50^e^	Myltenyi
	BV510	510	V7	**CD3**	OKT3	1/20^e^	Biolegend
	BV570	570	V8	**CD4**	RPA-T4	1/80^e^	Biolegend
	BV605	603	V10	**2B4**	C1.7	1/80^e^	Biolegend
	BV785	785	V15	**PD-1**	EH12.2H7	1/10^e^	Biolegend
488	FITC	520	B2	**CD57**	HNK-1	1/160^e^	Biolegend
	PerCPeFluor710	710	B10	**TCRgd**	B1.1	1/20^e^	Thermo Fisher
561	PE	576	YG1	**CD158**	HP-MA4	1/40^e^	Biolegend
	AlexaFluor594	617	YG3	**NKG2A**	131411	1/20^e^	Bio-Techne
	PE-Cy7	781	YG9	**CX3CR1**	2A9-1	1/20^e^	Biolegend
640	SparkNir685	685	R3	**CD19**	HIB19	1/80^e^	Biolegend
	APC Fire750 APC	787 660	R7 R1	**TCRgd2** **Streptavidin HLA_peptide_**	B6 /	1/160^e^ 1/11^e^	Biolegend BD Bioscience
/	/	/	/	**CD94**	HP-3D9	30 µg/mL	BD Bioscience
/	/	/	/	**Fc block**	Fc1.3216	1/100^e^	BD Bioscience

## Data Availability

Publicly available datasets were analyzed in this study.

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
