# Peer review of "Mapping and Characterization of HCMV-Specific Unconventional HLA-E-Restricted CD8 T Cell Populations and Associated NK and T Cell Responses Using HLA/Peptide Tetramers and Spectral Flow Cytometry"

_ijms, 2021, doi:10.3390/ijms23010263_

Round 1
Reviewer 1 Report
In this report, Charreau et al. describe a 20-color spectral flow cytometry analysis of PBMC samples from HCMV(+) and HCMV(-) blood donors (n=4/group), combining HLA/peptide tetramers and a panel of 18 mAbs specific for leukocyte differentiation markers. Phenotypic features of T (αβ and ɣδ) and NK cell subsets previously reported to be involved in the response to HCMV are described, with attention to HLA-E-restricted CD8+ T cells.
The results essentially mainly provide technical details of the analysis performed, relating their observations with previous studies. No significant differences between HCMV(+) and HCMV(-) are perceived for most parameters as expected given the small number of samples analyzed.
In summary, this study simply describes the use of a known refined flow cytometry technique of potential interest for monitoring the immune response to HCMV infection and vaccination, as concluded by the authors. Information provided in Materials and Methods (Samples and Reagents) on kidney transplant recipients is totally unrelated to the contents of this manuscript, presumably corresponding to other ongoing clinical studies that may eventually support its practical interest.
Author Response
Responses to the reviewer’s comments
Reviewer 1
In this report, Charreau et al. describe a 20-color spectral flow cytometry analysis of PBMC samples from HCMV(+) and HCMV(-) blood donors (n=4/group), combining HLA/peptide tetramers and a panel of 18 mAbs specific for leukocyte differentiation markers. Phenotypic features of T (αβ and ɣδ) and NK cell subsets previously reported to be involved in the response to HCMV are described, with attention to HLA-E-restricted CD8+ T cells.
The results essentially mainly provide technical details of the analysis performed, relating their observations with previous studies. No significant differences between HCMV(+) and HCMV(-) are perceived for most parameters as expected given the small number of samples analyzed.
In summary, this study simply describes the use of a known refined flow cytometry technique of potential interest for monitoring the immune response to HCMV infection and vaccination, as concluded by the authors. Information provided in Materials and Methods (Samples and Reagents) on kidney transplant recipients is totally unrelated to the contents of this manuscript, presumably corresponding to other ongoing clinical studies that may eventually support its practical interest.
Re : We thank the reviewer for these comments and for giving us the opportunity to revise and to improve our current manuscript. We acknowledge the fact that the present study mostly report on the development and validation of a methodological approach to investigate, in a single sample, both HCMV-specific CD8 T cell responses and other HCMV responses such as those mediated by subsets of NK and γδT cells. An original point was to perform the detection of HLA-E-restricted UL40-specific CD8 T in addition to the more conventional HLA-A-restricted CD8 T cell responses. Another feature of the study was to decipher these various immune responses using spectral flow cytometry which enables the use of a large panel of fluorophore/antibody pairs offering the opportunity to achieve deep immunophenotyping of a set of cell subsets. To set up the present protocol, we used blood samples that we previously tested for the absence or the presence of HCMV peptide-specific CD8 T cells using a set of pHLA tetramers and conventional flow cytometry. Four PBMC samples containing different HCMV peptide-specific CD8 T cells were then selected for the study. These 4 samples were issued from 2 HCMV+ healthy donors and from 2 HCMV+ kidney transplant recipients. The use of PBMC from kidney transplant recipients also provided the opportunity to detect and analyze cell subsets such as CD57+NKG2C+ NK cells which display higher frequency post-infection in patients. This point has now been clarified in the Materials and Methods section (page 12, lines 364-372).
Reviewer 2 Report
Please see attached document.

Author Response
Reviewer 2
The manuscript by Rousseliere et al describes methodology for analyzing the influence of human cytomegalovirus (HCMV) infection on the cellular immune system. Twenty-colour spectral flow cytometry including specific tetramers against conventional (HLA-A2-restricted) and unconventional (HLA-E-restricted) HCMV-specific T cells are used to identify CD8+ T cells specific for HCMV and further characterize their phenotype. General T cell populations are analyzed with the same approach as are T cell receptor (TCR) T cells and natural killer (NK) cells from HCMV-seropositive and -seronegative individuals. The main point of the manuscript is that the methodology developed by the authors can be used relatively routinely to analyze the influence of HCMV on the immune system and, thus, can be useful to better understand relationships between HCMV infection, the immune response, viral persistence, viral control, viral reactivation and other settings along the spectrum of HCMV infection status. It is clearly written and the figures are mostly easy to interpret. Only very minor editing for English is required. The initial and downstream gating strategies employed are appropriate and the authors generally demonstrate that with their approach, T and NK cell subset analysis down to the level of antigen-specific cells can be readily performed. However, very little data regarding the immunophenotyping of HCMV-specific T cells identified by tetramer staining is shown and there is little discussion of what the expression or absence of certain markers other than CCR7 and CD45RA might mean. It should primarily be viewed as a methods manuscript documenting the validity of the methods without generating new information on anti-HCMV immunity at this point. While this will be of value to the research community, I feel that some additional information could help clarify how their methodology provides an incremental step forward over commonly employed flow cytometry methods.
Re : We thank the reviewer for these comments and for giving us the opportunity to revise and to improve our current manuscript. We acknowledge the fact that the present study mostly report on the development and validation of a methodological approach to investigate, in a single sample, both HCMV-specific CD8 T cell responses and other HCMV responses such as those mediated by subsets of NK and γδT cells. An original point was to perform the detection of HLA-E-restricted UL40-specific CD8 T in addition to the more conventional HLA-A-restricted CD8 T cell responses. Another feature of the study was to decipher these various immune responses using spectral flow cytometry which enables the use of a large panel of 19 fluorophore/antibody pairs offering the opportunity to achieve deep immunophenotyping of a set of cell subsets. Nevertheless, we acknowledge that both the advantages of the methods and the selection of makers we used could be better introduced. Consequently, as you suggested, several changes have been introduced in the revised manuscript and they are presented below.
Minor comments
- While the advantages of spectral flow cytometry may be obvious to those more familiar with it, this reviewer is not so familiar and could benefit from more information on what differentiates it from conventional flow cytometry and why it is better than conventional 20 parameter flow cytometry.
Re : To answer this comment, we implemented the introduction with a better description of Spectral flow cytometry and a comparison with conventional flow cytometry, as suggested (Pages 2-3, lines 81-86)
- 2. The introduction refers to virus-specific effector NK cell and TCR T cell populations. I’m not convinced these populations have been shown to have specificity for HCMV in an antigenic
Re : Many thanks for this pertinent comment. We fully agree with the fact that in contrast to peptide-specific CD8T cell responses HCMV triggers no antigen-specific NK or gdT cells but rather induces the development of cell populations with particular phenotypic traits. Concerning NK cells, HCMV induces the emergence of NK cells expressing the CD94/NKG2C activating receptor and/or CD57, a marker for proliferation and cytotoxic activity.. Concerning gdT cells, previous studies have show that HCMV infection leads to an enhancement of γδT cell pool and more particularly of δ2neg γδT cells in particular in Kidney transplant recipients (Dechanet J. et al., J. Infect. Dis., 1999). Acording to your comment, our manuscript has been modified to introduce better the cellular events that associate with and can be considered as a signature of HCMV infection. Consequently, the sentence in the « Introduction» section has been corrected to avoid confusion (page 3, lines 90-91).
- The authors should state in the methods section or figure captions how many events are acquired for their analyses. In figure 1D, the T cells appear to be on a diagonal, suggesting possible compensation issues.
Re : For our study, a mean value of 40,000 events (alive lymphocytes)/samples were acquired for analysis. This point has now been introduced in the revised manuscript (pages13-14, lines 407-408). We confirm that the staining pattern showed in figure 1D is not a compensation issue. This special pattern of staining was obtained with PerCP-eFluor 710-anti-gdT mAb (clone B1.1, as indicated in the Table 1). Supporting this result, Park and colleagues in their published study establishing a large flow cytometry panel for immunophenotyping , obtained an identical staining profile using this B1.1 mAb clone labelled with PerCP-eFluor 710.( Park et al, 2020, Cytometry Part A, 97, 10 : 1044-1051; 10.1002/cyto.a.24213).
- Figure 1E shows the robust identification of HCMV antigen-specific CD8+ T cells. It is interesting that the HLA-E restricted cells seem to have a lower CD8 intensity than the HLA-A2-restricted T cells. Is there a precedent for this or is it a technical issue?
Re : This is indeed an interesting point and we confirm that low CD8 level was consistently observed for HLA-E-restricted CD8 T cells in our experiments. This is clearly not a technical issue since PBMC samples were concomittantly co-stained with HLA-E and HLA-A2 tetramers and mAb panel including anti-CD8 mAb. This feature was consistently observed in our past and current cohort studies both on HCMV+ kidney transplant recipients and on HCMV+ healthy controls thus we believe that low CD8 level is an hallmark of HLA-E CD8 T cells which may reflect a distinct activation capacity compared to « conventional » HLA-A-restricted CD8 T cell responses.
- With the 8 donors chosen, the authors report a reduction in the CD4+ T cell/CD8+ T cell ratio to around 1 in the HCMV+ group compared to ~2 in the HCMV- group. This is not something normally reported in the healthy adult population (except during primary infection) until older age. The age and sex of the donors in each group should be provided.
Re : Many thanks for this comment. The age and sex of the donors in each group is now provided in the « Materials and Methods » section (page 12, lines 364-372). The mean values are 47.5± 14.7 and 55± 14.7 years for HCMV- and HCMV+, respectively) and thus are not significantly different.
- 6. In figure 2A, it would be interesting to see the conventional and unconventional HCMV-specific CD8+ T cells compared. If they are all consistently TEMRA, the authors should comment on what this might mean and how it would be reflected in secondary markers to better demonstrate the discriminatory power of their methodology. The caption for 2C mentions immunophenotyping for NKG2C and NKG2A, but it is not shown on the plots. These both would be more appropriate for NK analysis and NKG2C for the TCR analysis. In general, a little more explanation for why the particular antibodies were chosen and what expression of the different markers represents would be helpful. What does the highly selective expression of KLRG1 on TEM and TEMRA suggest? On line 203, why is KLRG1 referred to as an “activating” receptor?
Re : As suggested, the Figure 2A (page 6) has been modified and now includes representative detection and characterisation for both HLA-E–restricted and HLA-A2-restricted anti-HCMV CD8 T cell populations. These data show that both CD8 T cell subsets display total or partial CD45RA+/CCR7- phenotype corresponding to TEMRA cells (see modified Figure 2A, page 6 and legend lines 180-190). This result is now commented in the text (page 5, lines173-176) as well as the context of the markers analyzed (page 6, lines 190-192 ; page 7, lines 197-202). The caption in figure 2C report on the markers used for costaining and shown in the figure and we confirm that they do include NKG2A and KNKG2C (labels on vertical axes). The rationale for the markers selected for our study is now improved in the text (page 7, lines 197-202). We acknowledge that KLRG1 is a lymphocyte co-inhibitory, or immune checkpoint, receptor expressed predominantly on late-differentiated effector and effector memory CD8+ T. This point has been corrected in the revised manuscript as suggested (page 7, line 205).
- 7. For analysis of TCR T cells, the authors use PBMC from HCMV+ and HCMV- kidney transplant patients without explaining why? Despite suggesting that HCMV infection leads to expansion of TCR T cells (as reported in some literature), there is no significant difference overall related to HCMV infection. Figure 3C might be clearer with a labelled right hand Y axis than with the double legend on the left hand Y axis.
Re : To set up the present protocol, we used blood samples that we previously tested for the absence or the presence of HCMV peptide-specific CD8 T cells using a set of pHLA tetramers and conventional flow cytometry. Four PBMC samples containing different HCMV peptide-specific CD8 T cells were then selected for the study. These 4 samples were issued from 2 HCMV+ healthy donors and from 2 HCMV+ kidney transplant recipients. This point has now been clarified in the Materials and Methods section (page 12, lines 364-372).
As suggested, Figure 3C has been modified to avoid confusion (page 8).
- For analysis of NK cells, it’s not clear why CD56- cells are included at the beginning. In the example shown for the healthy volunteer, there are so few NKG2C+ NK cells as to suggest the donor may be an NKG2C null individual. Again, an HCMV+ kidney transplant patient is used to analyze NK cells without any explanation. It has been reported previously that HCMV+ healthy controls have a higher frequency of NKG2C+ NK cells than HCMV- controls.
Re : Concerning NK cells, our gating strategy was to select NK on the basis of a CD3-/CD19-/CD56+/- phenotype in order to include all subsets of NK cells including those which express no or low CD56 level corresponding to terminally differentiated NK cells (Di Vito C.et al., Front. Immunol., 2019). We confirm that only low percentages of NKG2C+ cells were observed for NK from HCMV- while in contrast after HCMV infection (HCMV+) NKG2C+ NK cells frequency significantly increases in particular in KTRs as previously reported (Lopez-Verges S. et al., PNAS, 2011). This is not a technical issue since the same antibody was used for all the samples.
- Groups of 4 are small for statistical comparisons.
Re : We agree with this comment. However, the objective of this study was not to provide new data on HCMV infection but rather to establish and describe a robust protocol to study multiparameter events such as cell fequency and phenotype for various immune subsets in a single assay. This protocol was developed for the analysis of patient cohorts to decipher HCMV immune responses in transplanted recipients and to investigate the respective role of HLA-A2 versus HLA-E-restricted CD8 T cells in the homeostasis of the responses and of the control of infection. These studies are still ongoing.